# Nonlinear Effects Induced by Interactions among Functional Groups of Bacteria and Fungi Regulate the Priming Effect in Malagasy Soils

**DOI:** 10.3390/microorganisms11051106

**Published:** 2023-04-23

**Authors:** Benoît Jaillard, Kanto Razanamalala, Cyrille Violle, Laetitia Bernard

**Affiliations:** 1Eco & Sols, University Montpellier, IRD, INRAE, CIRAD, Institut Agro, 2 place Viala, 34060 Montpellier, France; 2Laboratoire des Radio-Isotopes, Antananarivo 101, Madagascar; 3CEFE, University Montpellier, CNRS, EPHE, IRD, 1919 Route de Mende, 34293 Montpellier, France

**Keywords:** assembly motif, bacteria, biodiversity, combinatorial analysis, ecosystem functioning, fungi, organic matter, priming effect, soil

## Abstract

The priming effect (PE) occurs when fresh organic matter (FOM) supplied to soil alters the rate of decomposition of older soil organic matter (SOM). The PE can be generated by different mechanisms driven by interactions between microorganisms with different live strategies and decomposition abilities. Among those, stoichiometric decomposition results from FOM decomposition, which induces the decomposition of SOM by the release of exoenzymes by FOM-decomposers. Nutrient mining results from the co-metabolism of energy-rich FOM with nutrient-rich SOM by SOM-decomposers. While existing statistical approaches enable measurement of the effect of community composition (linear effect) on the PE, the effect of interactions among co-occurring populations (non-linear effect) is more difficult to grasp. We compare a non-linear, clustering approach with a strictly linear approach to separately and comprehensively capture all linear and non-linear effects induced by soil microbial populations on the PE and to identify the species involved. We used an already published data set, acquired from two climatic transects of Madagascar Highlands, in which the high-throughput sequencing of soil samples was applied parallel to the analysis of the potential capacity of microbial communities to generate PE following a ^13^C-labeled wheat straw input. The linear and clustering approaches highlight two different aspects of the effects of microbial biodiversity on SOM decomposition. The comparison of the results enabled identification of bacterial and fungal families, and combinations of families, inducing either a linear, a non-linear, or no effect on PE after incubation. Bacterial families mainly favoured a PE proportional to their relative abundances in soil (linear effect). Inversely, fungal families induced strong non-linear effects resulting from interactions among them and with bacteria. Our findings suggest that bacteria support stoichiometric decomposition in the first days of incubation, while fungi support mainly the nutrient mining of soil’s organic matter several weeks after the beginning of incubation. Used together, the clustering and linear approaches therefore enable the estimation of the relative importance of linear effects related to microbial relative abundances, and non-linear effects related to interactions among microbial populations on soil properties. Both approaches also enable the identification of key microbial families that mainly regulate soil properties.

## 1. Introduction

The positive priming effect (PE), i.e., the acceleration of the soil’s organic matter (SOM) decomposition by the input of a fresh, easily decomposable, organic matter (FOM), is a key ecosystem process [1]. PE has long been explained by two major processes: (i) stoichiometric decomposition, which results from FOM decomposition, which induces the decomposition of SOM by the release of exoenzymes by FOM-decomposers; (ii) nutrient mining results from the co-metabolism of energy-rich FOM with nutrient-rich SOM by SOM-decomposers [2,3]. Energy-rich FOM can also come from organic matter desorbed by carboxylic acids released by FOM-decomposers [4]. The microbial actors and the biotic and abiotic determinants of the priming effect are therefore multiple [5]. All the proposed processes involve interactions among microbial communities, including competitive interactions among microbial populations for access to FOM and complementarity between the enzymatic activities of different microbial functional groups [6]. In order to improve the understanding of the priming effect, therefore, the identification of the microbial populations that favored each of the different processes is needed, either because of their own activities or because of their interactions with other soil microbial populations. DNA–SIP has been described as a direct method to identify microbial populations feeding on ^13^C-labeled organic matter [7]. However, as the PE generated by nutrient mining results from the co-metabolism between different sources of organic matter (labeled and unlabeled), such a method cannot discriminate the true FOM-decomposers implied in stoichiometric decomposition from the SOM-decomposers benefiting from labeled FOM catabolites and implied in “nutrient mining” [6]. Therefore, an appropriate statistical approach could help to better disentangle the involved processes.

Microbial biodiversity generates diverse effects on soil functions. The effects are most often decomposed into interaction effect, sensu stricto, and composition effect [8]. The composition effect reflects the specific activities of certain key species in the realization of a soil function and in the absence of interaction with other co-occurring species. It is therefore assumed to be cumulative and responds linearly to the presence of these species and their relative abundance within the community. The interaction effect gathers all positive and negative diversity effects other than the composition effect: it is therefore, by definition, non-linear. Several authors [8,9] separate the interaction effect from the composition effect on the basis of the linearity of the response of the composition effect to the relative abundance of key species [10]. The observed function of a community is compared with the sum of properties, characterizing all populations composing the community as they were grown in monoculture, weighted by their relative abundance. Using molecular methods, it is possible to discriminate between different populations of an assemblage and evaluate their relative abundance. However, the properties of each population grown in monoculture cannot be known because of the impossibility to isolate and cultivate many soil-native microorganisms. In this context, it is not possible to separate the two biodiversity effects on the basis of the linearity of the response of the composition effect to community biodiversity and species’ relative abundance using classical approaches.

Most statistical methods are based on the linearity of the response of a variable to be explained by a set of explanatory variables: variance analysis, linear regression, and principal component analysis. In the absence of interactions among species, a linear analysis of the property of a community explained by its biodiversity, especially species’ relative abundances, would therefore approximate the effect of community composition only. Conversely, a non-linear analysis of the community property would approximate the overall biodiversity effect of a community, i.e., the product of the composition effect and the interaction effect. Jaillard et al. [11,12] recently proposed an approach to explain ecosystem property by the co-occurrence of species in an ecosystem. This approach searches for species combinations that best account for observed variations in the ecosystem property. The approach enables the cluster of species into functional groups whose co-occurrence determines the ecosystem property. This clustering approach is non-linear. It can therefore be used to analyze the overall biodiversity effect of a microbial community. Compared with a linear approach, this non-linear approach allows for the inference of the effect of interactions among species on community properties.

Here, we aim to identify microbial taxa that predominantly contribute to the generation of PE in Malagasy Ferralsols, either by their own activities or through their interactions with other microbial taxa. Our approach is based on the comparison between linear and non-linear analyses of the same dataset. Three soil properties associated with SOM decomposition activity were analyzed: basal soil respiration, mineralization of ^13^C labeled wheat straw supplied to soil, and the PE generated by the straw supplied [13]. PE was calculated by removing the total carbon mineralized in the part resulting from wheat-straw and that which was registered in the non-amended subsample (i.e., basal respiration). These three soil properties were measured after seven and 42 days of incubation, because those times likely corresponded to different processes of PE generation. The composition of bacterial and fungal communities was assessed in pristine soils before incubation, by high throughput molecular analysis of 16S and 18S ribosomal genes, respectively. The gene sequences were assigned at the family level.

## 2. Materials and Methods

### 2.1. Dataset

The dataset used here was already partly published by [13]. Ferralsols from the Highlands of Madagascar were collected (May 2014, 57 sites) under herbaceous savannah dominated by the Poaceae *Aristida* sp., along gradients of temperature and rainfall. At each site, five soil cores were sampled from 0–10 cm depth and mixed to form a composite soil sample. The 57 composite soil samples were sieved to 2 mm, stored at 4 °C, then sent to the Genosol platform in Dijon, France. A 50 g aliquot of each composite soil sample was subsampled, then freeze-dried prior to genetic analysis. Soils have also been characterized as fully described in [13] and characteristics can be consulted in Appendix A.

Bacterial and fungal community compositions were assessed on 57 soil samples. Banks of 16S and 18S ribosomal sequences were prepared prior to pyrosequencing analyses as fully described in [14]. Pyrosequencing was carried out on a GS FLX Titanium (Roche 454 Sequencing System). Biocomputing analysis of the sequences was performed using the GnS-PIPE [15] as described in detail in Tardy et al. [16]. In order to compare the datasets efficiently and avoid biased community comparisons, the sample reads were reduced by random selection closed to the lowest datasets (3000 reads for 16S- and 18S-rRNA gene sequences, respectively, for each soil sample). The retained high-quality reads were used for taxonomy-based analysis using similarity approaches and dedicated reference databases from SILVA [17]. Assignments published in Razanamalala et al. [13] were made at the highest taxonomic level, i.e., the phylum level. However, in the present study, we preferred to perform assignments at a more functionally relevant level, i.e., the family level. Actually, we tested the class, order, and family phylogenetic levels, and family appeared as the best compromise to improve functional inference without drastically reducing the dataset due to the lack of known species and even genera in the SILVA database. In order to compare this work with that of Razanamalala et al. [13], the links between families and phyla are given in Appendix A. The relative abundance matrices had a dimension of *n* = 57 soils for *p* = 60 bacterial or fungal families, respectively. This number of families was chosen in order to be at the limit of the recommendations of the linear approaches in terms of the number of variables, which must be lower compared with the number of samples (see Section 2.2.3). Relative abundances per soil aliquot ranged from 1 to 1486, and from 5 to 1048, out of 3000 sequences, with a median of 4.0 and 11.5, for bacteria and fungi, respectively (Appendix A). The observation frequency of each family ranges from 32 to 100%, and 35 to 100% of soils, for bacteria and fungi, respectively (Appendix A).

In parallel, the soils were incubated in the absence or presence of ^13^C-enriched wheat straw (for details, see [13]). CO_2_ release was measured after seven and 42 days of incubation. Those two times were chosen because they likely corresponded to different processes of PE generation [13]. Three measurements were made: the CO_2_ released by soil in the absence of straw, the CO_2_ released by soil with straw supplied, and the ^13^C-labeled CO_2_ release due to the mineralization of the straw supplied, which was previously enriched with ^13^C. The measurements enabled the determination of three soil properties concerning SOM decomposition activity carried out by microbial communities: (i) basal soil respiration, equal to the release of CO_2_ by soil in absence of straw, (ii) straw mineralization, equal to the release of ^13^C-labeled CO_2_ due to the mineralization of the straw supplied only, and (iii) the priming effect (PE) generated by the straw supply, equal to the CO_2_ release by soil with straw supplied, minus the CO_2_ resulting from soil basal respiration and straw mineralization.

### 2.2. Linear Analyses

#### 2.2.1. Modeling SOM Decomposition by the Relative Abundance of Microbial Families

A multi-linear analysis without interactions assumes that each microbial family contributes to SOM decomposition in proportion to its relative abundance. The relative abundance of each family is then assumed to be independent of the relative abundance of other families. A multi-linear analysis therefore models SOM decomposition as a linear combination of the relative abundances of families that co-occur in soil. The model consisted of weighing the relative abundance of each family to optimize the fit to SOM decomposition.

#### 2.2.2. Evaluation of the Model Quality

The explanatory power of the model was measured by its coefficient of determination, R^2^, that is the ratio of the variance explained by multi-linear analysis on the total observed variance. The model quality, i.e., the compromise between the model precision and the number of explanatory variables, hereafter, the number of families (plus one for the regression intercept) was measured by its Akaike Information Criterion AICc, corrected for small sample sets [18].

#### 2.2.3. Model Reduction by Variable Selection

A multi-linear analysis is improved by reducing the number of explanatory variables. We used a backward stepwise variable selection, based on the progressive removal of microbial families considered as irrelevant explanatory variables because of non-significant contributions to SOM decomposition. Each family was successively removed from the set of explanatory variables, and the effect of each of these removals on model quality was evaluated. The family whose removal maximized the model quality was considered irrelevant and was definitively removed from the set of explanatory variables. We repeated the procedure while the removal of a family increased the model quality. We stopped when removing any remaining families decreased model quality. This means that all remaining families were then identified as relevant explanatory variables, because their contribution to SOM decomposition was significant and because their inclusion in the set of explanatory variables improved model quality. Throughout the variable selection procedure, the quality of the model was measured by its AICc.

A linear model consists of a system of *n* equations with *p* unknowns, hereafter, *n* observed soils for *p* microbial families: it requires a non-zero number of degrees of freedom, i.e., a number *n* of observations greater than the number *p* of families, plus one (*n* > *p* + 1). The bacteria and fungus relative abundance matrices comprised *n* = 57 soils for *p* = 60 families: this mathematical constraint imposed to keep only *p* = 55 families for *n* = 57 soils. When analyzing only the bacterial families, or only the fungal families, we therefore removed the five least abundant families from the bacteria and fungi relative abundance matrices, respectively.

### 2.3. Clustering Analysis

#### 2.3.1. Modeling SOM Decomposition by the Co-Occurring of Relative Abundance Classes of Microbial Families

A clustering analysis is based on the occurrence of families in microbial communities, i.e., qualitative data. The presence of each microbial family in soil, however, was measured by their relative abundance, i.e., quantitative data. The relative abundance of each family was therefore converted into classes of relative abundance, then each quantitative matrix of relative abundance was converted into a qualitative one-shot encoding table. Clustering methods are all the more effective when each family to be clustered is equitably observed. The least frequent bacterial and fungal families were observed in 32% and 35% of sampled soils, respectively. The distribution of relative abundance of each family was therefore segmented into classes of relative abundance, centered on an observation frequency of 30% (Appendix A). The ubiquitous families, i.e., the families observed in 100% of soils, were segmented into three classes of relative abundance, noted 1/3, 2/3, and 3/3 from the least to the most abundant classes. The least frequent families were not segmented and were noted 1/1 as a reminder (Appendix A). Families in between were segmented into two classes and noted as 1/2 and 2/2. The segmentation thresholds were determined in such a way that the numbers of observations in each class were equal to, within one unit. The resulting one-hot encoding tables contained 153 and 175 classes of family relative abundance, with an average observation frequency of 31.7% and 32.4% (median 32.2% and 33.3%) for bacteria and fungi, respectively (Appendix A). The clustering analysis then aimed to group classes of family relative abundance whose co-occurrence best accounted for variations of SOM decomposition. The method then looked to maximize the inter-group variance of SOM decomposition associated with different combinations of functional groups, i.e., the different assembly motifs [11]. As in the linear modelling, the explanatory power of the model was measured by the coefficient of determination, R^2^, that is the ratio of the variance explained by the clustering analysis on the total observed variance.

#### 2.3.2. Model Reduction by Variable Selection

As a linear model, a clustering model is improved by reducing the number of explanatory variables. We used a backward stepwise selection similar to the one applied to linear analysis. The first difference was that each family was segmented into several classes of relative abundance: removing a family here therefore meant removing all the classes of relative abundance from the family. A second difference is that a clustering model enables the clustering of families into functional groups and observations into assembly motifs, i.e., combinations of functional groups: we imposed that for a family to be considered as an irrelevant variable, its removal must change neither the functional groups nor the assembly motifs. A third difference is that all the least significant classes of family relative abundance are clustered in the same functional group by default, with the residual group labeled by the letter of highest rank: only the removal of families belonging to this group are therefore tested.

#### 2.3.3. Hierarchy of Functional Groups

A clustering analysis ultimately provided a hierarchical tree of the functional groups of classes of family relative abundance that best accounted for SOM decomposition. The functional groups are sorted from left to right, from those that best to least explain SOM decomposition. Within each functional group, the classes of family relative abundance were also sorted from left to right, based on their ability to explain SOM decomposition. The residual group that clusters the least significant classes of family relative abundance is therefore on the right, labeled by the letter of highest rank, and the least significant classes of family relative abundance belonging to this group is on the right of this functional group.

#### 2.3.4. Evaluation of the Model Quality

As for a multi-linear model, the explanatory capacity of a clustering model was measured by its coefficient of determination, R^2^, that is the ratio of variance explained by the clustering analysis on the total observed variance. The quality of the model was measured by its AICc, i.e., the trade-off between model precision and the number of explanatory variables, hereafter, the number of families were considered relevant. As explained above, the variable selection procedure was based on constant numbers of functional groups and observed assembly motifs. However, the number of families on which the clustering analysis was based decreased. The number of variables taken into account in the AICc calculation were the family numbers, instead of the numbers of observed assemblage motifs that stayed constant [11].

#### 2.3.5. Linear Effects Associated with Each Functional Group

The clustering analysis looks to maximize the inter-group variance of SOM decomposition associated with the different assembly motifs, which are the combinations of functional groups. It is also possible to assess the linear effects explained by clustering analysis using a variance analysis of SOM decomposition explained by each functional group of classes of family relative abundance and their statistical interactions.

### 2.4. Statistical Computations

All computations were performed using R-software [19]. Linear and multi-linear regressions were performed using the lm function, and variance analyses using the aov function of the R-package stats. Clustering analyses were performed using the fclust function of the R-package functClust. Backward stepwise selections were performed using the lm_step and fclust_step functions. Comparison of medians was performed using a non-parametric Kruskal–Wallis test (function kruskal.test of the R-package stats). All statistical tests were performed at the probability threshold of *p* < 0.001. All scripts are available on the INRAE dataverse at https://doi.org/10.15454/LRY3M2 (accessed on 20 March 2023).

## 3. Results

### 3.1. Linear Analyses of the Relationship between SOM Decomposition Activity and Environmental Conditions of Sampling Sites

Basal soil respirations after seven and 42 days of incubation were positively correlated (R^2^ = 0.773; *p* < 10^−10^). Straw mineralization after 42 days did not depend on that observed after seven days of incubation (R^2^ = 0.007; *p* = 0.238), and the priming effects after seven and 42 days of incubation were negatively correlated (R^2^ = 0.203; *p* = 0.256 × 10^−3^). This means that soils with a high PE after seven days had a low PE after 42 days of incubation.

In general, SOM decomposition functions did not vary with precipitation at sampling sites. In contrast, basal soil respirations after seven and 42 days were negatively correlated (R^2^ = 0.496; *p* = 0.601 × 10^−9^ and R^2^ = 0.391; *p* = 0.118 × 10^−7^, respectively) with temperatures, and PE after seven and 42 days were negatively and positively correlated, respectively (R^2^ = 0.043; *p* = 2.064 × 10^−5^; and R^2^ = 0.180; *p* = 0.588 × 10^−3^), with temperatures at the sample sites.

### 3.2. Multi-Linear and Clustering Analyses of the Relationship between Initial Composition of Bacterial and Fungal Communities and SOM Decomposition Activity

Both multi-linear and clustering approaches to the relative abundance of bacterial families accurately explained SOM decomposition functions, i.e., basal soil respiration, straw mineralization, and the priming effect, after seven and 42 days of incubation (Table 1). The six multi-linear models selected, in median, 26 key families out of 55, with model degrees of freedom of 28 and R^2^ of 0.959. The Fisher ratios were around 25.0, all associated with a probability lower than 10^−10^. The median AICc of the six models was −316.6. The clustering analyses selected, in median, only 14 key families out of 60, with model degrees of freedom of 46 and R^2^ of 0.965. The Fisher ratio equaled 49.3 and the AICc −356.2. Regarding fungal communities (Table 1), the variable selection procedure applied in the multi-linear and clustering models was almost as efficient as for bacteria communities, with 18 and 16 selected key families, respectively. While the median R^2^ of the multi-linear models were around 0.632, those of clustering models reached 0.933, indicating the same accuracy for fungal communities as bacterial communities.

The variable selection procedure retained, in median, only 14 key families out of 120, for the six models (Table 1). The number of degrees of freedom was thus 106, the coefficient of determination R^2^ 0.954, and the F-ratio 54.3. All F-ratio were associated with a probability lower than 10^−10^. The resulting AICc was −377.0 in median, i.e., the best mean AICc among all models. Applied to soil bacteria and fungus communities combined together, the clustering analyses provided parsimonious and accurate models.

### 3.3. Identifying Bacterial and Fungal Families That Regulate SOM Decomposition Activity

Both multi-linear and clustering approaches enable the identification of bacterial or fungal families as the key actors in basal soil respiration (Appendix A), straw mineralization (Appendix A), and the priming effect (Appendix A) after seven and 42 days of incubation, by separately analyzing each of the two communities. The results are summarized in Table 2.

For each SOM decomposition function and time of incubation, families identified as key actors in initial soil communities by either the multi-linear or the clustering approach accounted for 47% and 42% of the total number of bacterial and fungal families, respectively. Only 13% and 7% of bacterial and fungal families were commonly identified as key for using both approaches, while 37% and 51% were commonly rejected as non-significant (Table 2). This indicates that multi-linear and clustering analyses modeled different aspects of the relationship between microbial families of initial soil communities and SOM decomposition, and this difference was stronger for fungi than for bacteria.

The clustering analyses of bacterial and fungal families gathered together in the same dataset confirmed the results (Appendix A, summarized in Table 3). The variable selection procedure retained less than 14% of total families for a given SOM decomposition function and a given incubation duration, and also rejected 44% of total families showing no relationship for any SOM decomposition function at any incubation duration. This indicates that a small number of microbial families of initial soil communities explained a large part of SOM decomposition after seven and 42 days of incubation.

Although each family was observed in about 30% of soils, the extent of the relative abundances of bacterial and fungal families were very broad, ranging from 1 to 1486, and from 5 to 1048 out of 3000 sequences, respectively (Table 2 and Table 3, Appendix A). However, a comparison of medians confirmed that key families were not related to their relative abundance, whatever the SOM decomposition function, the time of duration, or the modelling approach (Appendix A).

### 3.4. Identity of Potentially Key Bacterial and Fungal Controlling the Priming Effect

The main results of the clustering analyses of bacterial and fungal families gathered together in the same dataset are reported in Figure 1, Figure 2 and Figure 3. Figure 1 shows the clustering of relative abundance classes of the microbial families of initial soil communities identified as key for basal soil respiration (Figure 1a,b, see Appendix A) and straw mineralization (Figure 1c,d, see Appendix A) after seven and 42 days of incubation. Figure 2 and Figure 3 report the clustering analyses of key bacterial and fungal families for the priming effect after seven and 42 days of incubation, respectively (see Appendix A).

Ten bacterial and nine fungal families of initial soil communities were identified as key for the priming effect after seven days of incubation. The 19 microbial families were clustered into five functional groups (Figure 2). Functional groups were labeled from A to E by their decreasing contribution to PE (Figure 2a). Within each group, the relative abundance classes of microbial families are sorted from left to right, by their decreasing contribution to PE (Figure 2a). The specific contribution of the relative abundance classes of family ranged from −15% to +20% of the median PE (Figure 2b). Some families showed a linear contribution, such as the bacteria *Caulobacteraceae*, while others displayed a non-linear contribution, such as the fungus *Sarcosomataceae* and the bacteria *Bacillaceae* (Figure 2b). The first main functional group, noted A, included three bacterial families, *Caulobacteraceae* in its highest relative abundance (class 3/3), *Methylocystaceae* in its lowest relative abundance (class 1/2), and *Bacillaceae* in its medium relative abundance (class 2/3), and three fungal families, *Sarcosomataceae* (class 2/3), *Pluteaceae* (class 2/3), and *Teratosphaeriaceae* (class 3/3) (Figure 2a). The functional group B contained two bacterial families, *Phyllobacteriaceae* (class 1/1) and *Acidobacteria*-Gp5 (class 2/3), and the fungal family *Ambisporaceae* (class 2/3) (Figure 2a). A variance analysis indicates that the functional groups A and B are associated with a PE that is 39% and 17% higher on average, respectively, than the median priming effect (P(A) < 10^−10^; P(B) = 0.63 × 10^−9^). Moreover, their statistical interaction is associated with an even higher PE of 23% (P(A:B) = 0.51 × 10^−6^). As a corollary, all assembly motifs containing the functional group A were associated with high PE, and the assembly motifs containing both the functional groups A and B were associated with the highest PE (Figure 2c). At the opposite, the functional groups C and D are associated with a PE of an average 14% and 12%, respectively (P(C) = 0.37 × 10^−6^; P(D) = 0.53 × 10^−6^), lower than the median priming effect. However, the functional group C interacts positively with the A and B groups and is associated with a PE of 20% and 3%, respectively (P(A:C) = 0.44 × 10^−7^; P(B:C) = 0.62 × 10^−3^), higher than the median. The functional group C clustered three bacterial families, *Planococcaceae* (class 1/2), *Acidobacteria*-Gp3 (class 2/2), and *Xanthobacteraceae* (class 2/2), and two fungal families, *Boletaceae* (class 1/3) and *Tubeufiaceae* (class 2/3). The functional group D included two bacterial families, *Acidobacteria*-Gp2 (class 1/3) and *Solirubrobacteraceae* (class 2/3), and the fungal family *Cordycipitaceae* (class 1/3): it did not interact with other functional groups. The last functional group E gathered all other bacterial and fungal families identified as key: it is not associated with any significant variation in the priming effect. Finally, the clustering of 19 families of initial soil communities identified as key accurately explained the priming effect after seven days of incubation (R^2^ = 0.959; AICc = −428.1; *p* < 10^−10^) (Figure 2d). The clustering was related neither to precipitation (*p* = 0.503) nor to temperature (*p* = 0.027) at sampling sites.

The priming effect intensity was lower after 42 days than seven days of incubation. Seven bacterial and six fungal families of initial soil communities were identified as key for the PE. The 13 microbial families were clustered into five functional groups (Figure 3). The specific effect of relative abundance classes of family ranged from −43% to +60% of the median PE (Figure 3b). The main first functional group A included the group of unknown bacteria (class 3/3), two bacterial families, *Polyangiaceae* (class 2/3) and *Actinospicaceae* (class 2/3), and three fungal families, *Ajellomycetaceae* (class 3/3), *Phaeosphaeriaceae* (class 2/3), and *Montagnulaceae* (class 1/3): the group A is associated with a PE 140% higher than the median (P(A) < 10^−10^) (Figure 3a). This means that the co-occurrence of at least two families belonging to the group A doubled the PE variation associated with only a single family (Figure 3b); all assembly motifs containing the functional group A were therefore associated with high PE (Figure 3c). The functional group B is also associated with a PE 51% higher than the median (P(B) = 0.85 × 10^−9^); yet it contained only one fungal family, *Chaetomiaceae,* in its lowest relative abundance (Figure 3a). The functional group C clustered the bacterial families *Burkholderiaceae* (class 3/3), *Phyllobacteriaceae* (class 1/1), *Sinobacteraceae* (class 2/2), and *Rhodospirillaceae* (class 2/3), and the fungal family *Montagnulaceae* (class 3/3) (Figure 3a). This group is associated with a PE 86% lower than the median (P(C) < 10^−10^). It strongly interacted with the functional group B, associated with a PE 48% even lower than the median (P(B:C) = 0.42 × 10^−3^). As a consequence, the assembly motifs containing the functional group C, and even more when associated with B, showed the lowest PE (Figure 3c). In addition, functional group D is associated with no specific variation of PE, but it strongly interacted with A, which is associated with a PE 63% higher than the median (P(A:D) = 0.34 × 10^−3^): it included the fungal family *Ajellomycetaceae* (class 2/3) and the bacterial family *Burkholderiaceae* (class 1/3) (Figure 3a). Finally, the clustering of only 13 families identified as key in initial soil communities accurately and parsimoniously modeled the PE after 42 days of incubation (R^2^ = 0.933; AICc = −395.7; *p* < 10^−10^) (Figure 3d). The clustering was related neither to precipitation (*p* = 0.148) nor to temperature (*p* = 0.003) at sample sites.

Except for the bacterial family *Phyllobacteriaceae* and the fungal family *Cordycipitaceae*, all families identified as key in initial soil communities for the priming effect were different after seven and 42 days of incubation (Figure 2b and Figure 3b).

## 4. Discussion

### 4.1. A Few Microbial Families Were Associated with Significant Variations in SOM Decomposition Functions

A striking result is that whatever the soil function, a small number of microbial families of initial soil communities explain most of the SOM decomposition functions after seven and 42 days of incubation. When analyzing bacterial and fungal families gathered together, the clustering analysis identified less than sixteen families as key out of 120, i.e., less than 14% of observed families. These findings confirm that, for a given ecosystem property, biodiversity is most often a saturating function; only a fraction of the identified species contribute significantly to the emergence of the property studied [20,21,22]. They highlight the importance of procedures of variable selection for analyzing the relationships between biodiversity and ecosystem functioning; these procedures reduce the communities to only the species that are key, at one time or another, for the property studied [23,24]. Note that our analyses are based on microbial composition of initial soil communities before incubation; the microbial communities obviously changed without our knowledge during incubation, mainly by changing their relative abundances. The initial composition of soil microbial communities is therefore key in anticipating SOM decomposition functions after seven and 42 days of incubation, especially the priming effect.

Another striking result is that when bacterial and fungal families were analyzed together, all functional groups contained both bacteria and fungi. This means that some bacteria and fungi are functionally redundant. This functional redundancy suggests that, despite their morphological and physiological differences, bacteria and fungi may be associated with similar processes. However, it is possible that this functional redundancy is only a façade. Like any statistical analysis, a clustering analysis is not causal; it only relates a variable to be explained with explanatory variables. The clustering of families or classes of families in relative abundance in the same functional group therefore does not prejudge the role played by each of the clustered families. It only indicates that different combinations of microbial families are associated with different variations in SOM decomposition activity which, moreover, take place seven and 42 days after analysis of the initial composition of soil microbial communities. The functional groups generated by clustering analysis can therefore be composed of microbial families that directly and actively affect SOM decomposition activity after seven or 42 days of incubation, and families that contribute to variations in biotic or abiotic conditions are associated with variations in SOM decomposition.

For instance, the bacterial *Methylocystaceae* family is identified as key in the initial soil community for basal soil respiration, straw mineralization, and the priming effect after seven days of incubation. As the *Methylocystaceae* family is a strictly methanotrophic family that releases CO_2_ by oxidizing CH_4_ [25], they could not have contributed to any heterotrophic respiration. Therefore, our results highlight that soil conditions favorable to the *Methylocystaceae* family promote the priming effect after seven days of incubation. As another example, the bacterial *Polyangiaceae* family is identified as key in the initial soil community for basal soil respiration and the priming effect after 42 days of incubation; it then belongs to the functional group A, which is associated with a high priming effect. *Polyangiaceae* are bacteria that prey on other bacteria [26]; this family possibly plays a positive role in regulating specific populations having a negative effect on PE.

Our results also show that key families are neither the most frequent nor the most abundant families. For instance, when the functional groups A and B co-occurred in the initial soil community, the highest PE was observed after seven days of incubation. All bacterial families identified, *Caulobacteraceae*, *Methylocystaceae,* and *Bacillaceae,* in functional group A, and *Phyllobacteriaceae* and *Acidobacteria*-Gp5 in functional group B, are not highly abundant. Likewise, the fungal families *Sarcosomataceae* and *Pluteaceae* in functional group A, and *Ambisporaceae* in functional group B, are not very abundant. Only the ubiquist fungus *Teratosphaeriaceae* was moderately abundant and co-occurred in its highest relative abundance class in its initial soil community. All the microbial families were, however, associated with high PE. When the functional groups A and B co-occurred in the initial soil community, the highest PE was observed after 42 days of incubation. The unknown bacteria were very abundant and co-occurred in their most abundant relative class, although they correspond to a phylogenetically and functionally heterogeneous group. The fungus families *Phaeosphaeriaceae* and *Ajellomycetaceae*, however, were much less abundant in soils and co-occurred in the medium relative abundance class. It is possible that the most abundant families play an active role in SOM mineralization functions after seven or 42 days of incubation, while the less abundant families were rather associated with biotic or abiotic conditions favoring SOM decomposition.

We should, however, remember that (i) first, next generation sequencing technology is not a quantitative molecular method [14]; (ii) second, the occurrence of each microbial family was determined by only a few sequences, although it was observed in about 30% of soils. This lack of relationship between observed effects and frequency or abundance of microbial families should nevertheless be a question to us. In ecology, it is common to eliminate the least frequent or least abundant families because these rare species are a priori assumed to be inefficient [27,28]. This practice tends to underestimate the contribution of rare families to community functioning, even though some of them may play a key role [29,30,31]. In bacterial communities, the clustering analysis confirmed that most of the families were eliminated in the multi-linear analysis because of their low frequency or abundance not being key. In fungal communities, clustering analysis identified more than half of the families as key that were eliminated in the multi-linear analysis due to their low frequency or abundance. Our results thus highlight that taxonomic rarity is not a relevant criterion. Clustering analysis enables us to more deeply explore the relationship between biodiversity and ecosystem functioning than multi-linear analysis. It should be favored in looking for species that play a key role in the relationships between biodiversity and ecosystem functioning.

### 4.2. Multi-Linear and Clustering Analyses Enable the Identification of Microorganisms That Stimulate SOM Decomposition Activities after Seven and 42 Days of Incubation

Our results show that SOM decomposition activities are accurately explained by variations in the relative abundance of bacterial families in pristine soil. It is more accurately and parsimoniously explained by the co-occurrence of bacterial families at different levels of relative abundance. However, the improvement is marginal. Multi-linear and combinatorial approaches are therefore equally relevant. These results support the observations of Razanamalala et al. [13] made at the phylum level on the same samples, who found that bacteria were strongly linearly correlated with PE. The effect induced by bacteria on SOM decomposition activity would therefore be close to a composition effect.

Contrastingly, the relationship between fungi occurrence and any SOM decomposition activity at any time duration are hardly explained by multi-linear analyses, in contrast to the clustering analysis, which remains accurate. The relationships between fungi occurrence and SOM decomposition activities are thus non-linear; their occurrence moderately explains the PE after seven days of incubation, but much more explains the PE after 42 days. At this time, the occurrence of some functional groups composed of both bacteria and fungi in initial soil communities is related to a double PE, while the occurrence of other functional groups is related to significantly reduced PE. The main bacterial and fungal families selected for their potential involvement in PE recorded after 42 days of incubation are not the same as at the beginning of incubation, and non-linear processes of high intensity have taken over linear processes.

### 4.3. Linear and Clustering Analyses Enlighten the Priming Effect

Several authors have suggested that mechanisms underlying the priming effect can occur simultaneously: one may dominate for a period of time and then be replaced by another depending on changes in environmental conditions, including the availability of mineral nitrogen in soil solution [2,32,33]. Based on the literature, Razanamalala et al. [13] proposed that the PE measured after seven days was mainly generated by stoichiometric decomposition, and that the PE measured after 42 days was mainly the result of nutrient mining. Strikingly, we showed that microbial families favoring PE measured after seven and 42 days of incubation were not the same.

After seven days of incubation, the functional group A associated with the highest priming effect seems functionally convergent as it is mainly composed of bacterial families endowed with specific enzymatic capacities for the decomposition of lignin (*Caulobacteraceae,* [34]) and cellulose (*Bacillaceae*, [35]). This group also contains saprotrophic fungi well known for their lignocellulolytic capacities [36]. These results support the stoichiometric decomposition hypothesis after seven days of incubation, yet none of these families appeared to have a preponderant role, either in basal soil respiration or in straw mineralization. Although they might play a key role in PE, they are not among the most abundant nor the most frequent families. These families could therefore benefit from the exoenzymatic activity of other populations directly involved in straw mineralization as suggested in the stoichiometric hypothesis. Our results also show that PE depended on positive and negative interactions between families of group A and other functional groups. For instance, group B, which increased the effect of group A, is composed of two bacterial families’ characteristics of rich media, the *Phyllobacteriaceae* [37], or at least enriched in nutrients, the *Acidobacteria*-GP5 [38], and whose common feature is their ability to denitrify. Group C has a rather negative effect on PE as well as on group A, and is functionally more diversified than groups A and B.

When the labile part of FOM and the available nitrogen is exhausted, SOM-decomposers can use the energy of FOM catabolites to mine nitrogen in SOM [5]. Fontaine et al. [36] suggested that fungi are the main decomposers of cellulose and are thus the main actors of the nutrient mining PE. This important role in nutrient mining is also linked to the ability of fungi to explore a large volume of soil, having their mycelium on FOM and SOM simultaneously. It is indeed reasonable to assume that interactions between bacteria are limited due to their relative immobility. In contrast, fungi develop an extensive mycelial network that allows them to interact over long distances with other populations of bacteria and fungi. After 42 days of incubation, unlike after seven days of incubation, the functional group A that increased PE by 140% is functionally quite heterogeneous. It is composed of two families of saprotrophic Ascomycetes (*Phaeosphaeriaceae* and *Ajellomycetaceae*, [39,40]), a family of acidophilic Actinomycetes (*Actinospicaceae*, [41]) known for their ability to produce secondary metabolites [42], and unknown bacteria, which are functionally heterogenous but generally considered oligotrophic because they do not grow in the too-rich conditions of laboratory environments. Against all expectations, it is the bacteria (unknown and *Actinospicaceae*) that seem to engage in co-metabolism between FOM and SOM, not the fungi. However, as the group of unknown bacteria brings together various families under the same name, it can actually be divided into families specialized in either the breakdown of FOM or SOM, without co-metabolism. Enrichment of the databases should gradually reduce the size of this group in future studies. It has recently been suggested that PE could also be generated by the release of molecules with a high affinity for soil minerals, making it possible to desorb the initially associated labile organic matter rich in N and P [5]. Representatives of *Burkholderiaceae* are able to weather minerals by the release of gluconic acid [43]. However, in our study, *Burkholderiaceae* have a strong inhibitory role on PE after 42 days of incubation, linked to a density effect (their lowest and highest abundance classes are found in functional groups C and D). At the same time, they have an inhibiting role in straw mineralization. *Burkholderiaceae* have other capacities, such as exopolysaccharide production, antifungal release, and organic and inorganic *p* solubilization via siderophores production [44,45]. All those capacities can interfere directly or indirectly in the PE generation process. Therefore, the role of this family in the late priming effect has to be more deeply investigated in the future.

## 5. Conclusions

The clustering approach makes it possible to highlight taxa with non-linear effects on a flow resulting from complex processes, such as the priming effect, but also the effect on this flow of interactions between certain taxa. This approach has the advantage of not being constrained by the ratio between the number of variables and the number of samples, unlike linear statistical approaches. Normally applied to occurrence matrices, it has been adapted here to an abundance matrix by dividing each taxon into one, two or three abundance classes according to their respective densities. The results obtained are surprising and open up new challenges for their interpretation. This approach has proven its interest and deserves to be applied to more recent datasets resulting from sequencing techniques and more advanced bioinformatics analysis, which could only improve the quality and precision of the results. It is obvious that the biofunctioning of the soil is subservient to interactions between various functional entities and that this type of method considerably increases the field of possibilities in terms of understanding its various workings.

## Figures and Tables

**Figure 1 microorganisms-11-01106-f001:**
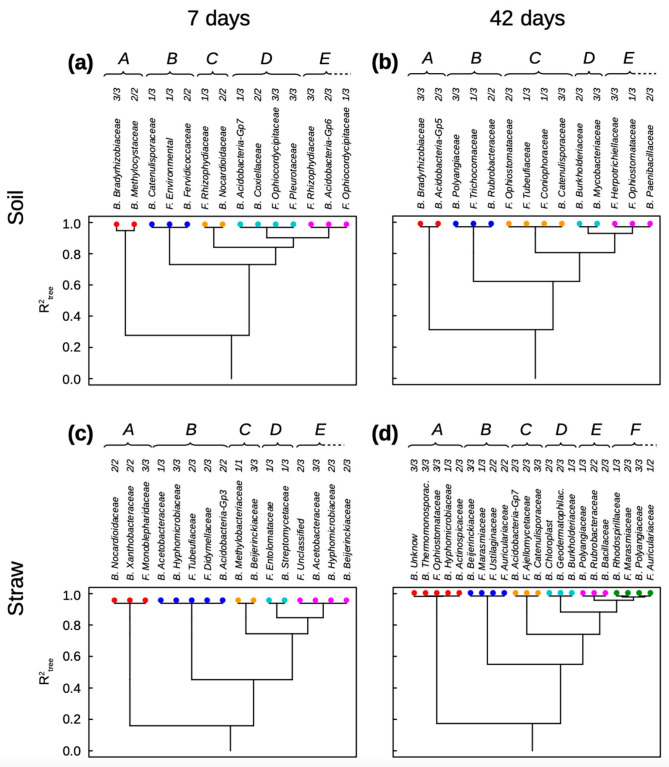
Clustering analysis of basal soil respiration (**a**,**b**) and straw mineralization (**c**,**d**) after seven (**a**,**c**) and 42 days of incubation, explained by the relative abundance classes of bacterial and fungal families in soil. The functional groups and the classes of family relative abundance inside each functional group are sorted from left to right by their decreasing effects on the soil property. Each functional group has a color attributed: red for A, dark blue for B, yellow for C, light blue for D, pink for E, and green for F.

**Figure 2 microorganisms-11-01106-f002:**
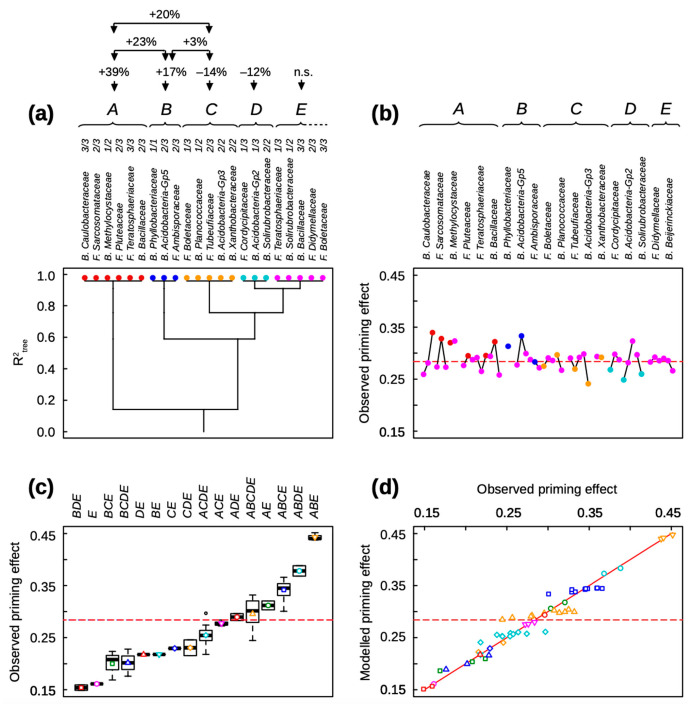
Clustering analysis of the priming effect after seven days of incubation, explained by the relative abundance classes of bacterial and fungal families in soil. (**a**) Clustering tree of classes of family relative abundance. The functional groups and the classes of family relative abundance inside each functional group are sorted from left to right by their decreasing effects on the priming effect. Each functional group has a color attributed: red for A, dark blue for B, yellow for C, light blue for D, pink for E. The above values indicate the significant effects (at *p* < 0.001) of the different functional groups and their interactions. (**b**) Mean observed priming effect of soils containing a given class of family relative abundance. The linked points correspond to the different classes of increasing relative abundance (from left to right) for each family. The used symbols are the same as those used in (**a**). (**c**) Boxplots of the observed priming effect sorted by assembly motifs, i.e., combinations of functional groups. The used symbols are the same as those used in (**b**). (**d**) Modeled versus observed priming effect. Different symbols correspond to different assembly motifs, i.e., combinations of functional groups. The red line is the bisector. (**b**–**d**) The dotted line is the mean observed priming effect.

**Figure 3 microorganisms-11-01106-f003:**
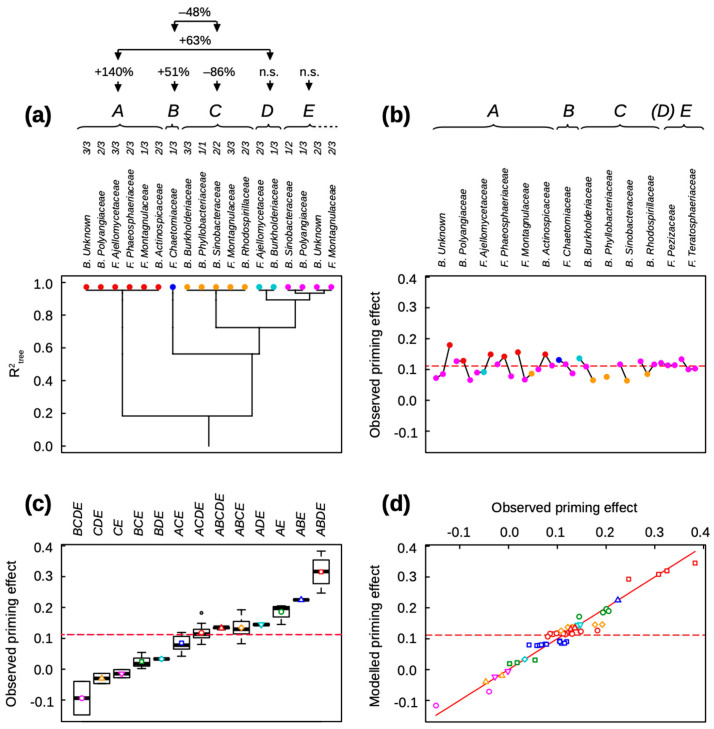
Clustering analysis of the priming effect after 42 days of incubation, explained by the relative abundance classes of bacterial and fungal families in soil. (**a**) Clustering tree of classes of family relative abundance. The functional groups and the classes of family relative abundance inside each functional group are sorted from left to right by their decreasing effects on the priming effect. The above values indicate the significant effects (at *p* < 0.001) of the different functional groups and their interactions. (**b**) Mean observed priming effect of soils containing a given class of family relative abundance. The linked points correspond to the different classes of increasing relative abundance (from left to right) for each family. The used symbols are the same as those used in (**a**). (**c**) Boxplots of the observed priming effect sorted by assembly motifs, i.e., combinations of functional groups. The used symbols are the same as those used in (**b**). (**d**) Modeled versus observed priming effect. Different symbols correspond to different assembly motifs, i.e., combinations of functional groups. The red line is the bisector. (**b**–**d**) The dotted line is the mean observed priming effect.

**Table 1 microorganisms-11-01106-t001:** Statistics of multi-linear and clustering analyses of the relationships between relative abundance of bacterial and fungal families and basal soil respiration, straw mineralization, and the priming effect. Analyses of relative abundance of bacteria, fungi, and bacteria and fungi get together.

	Mineralization	Duration ofIncubation	Statistical Model	Initial Number of Families	Number of Key Families	Degrees of Freedom	R^2^	F-Ratio	AICc
bacteria	soil	7 days	lm	55	32	22	0.993	112.1	−319.1
			fclust	60	17	43	0.966	39.3	−328.2
		42 days	lm	55	28	26	0.981	50.8	−304.9
			fclust	60	12	48	0.966	84.6	−354.8
	straw	7 days	lm	55	38	16	0.985	30.4	−91.3
			fclust	60	13	47	0.963	59.2	−246.9
		42 days	lm	55	19	35	0.883	14.6	−314.0
			fclust	60	15	45	0.927	24.9	−357.5
	priming effect	7 days	lm	55	22	32	0.879	11.2	−351.6
			fclust	60	20	40	0.985	99.0	−480.5
		42 days	lm	55	24	30	0.936	19.6	−347.8
			fclust	60	13	47	0.939	32.5	−401.0
fungi	soil	7 days	lm	55	11	43	0.598	6.1	−209.3
			fclust	60	26	34	0.977	30.7	−302.4
		42 days	lm	55	23	31	0.666	4.3	−193.4
			fclust	60	14	46	0.929	18.2	−307.1
	straw	7 days	lm	55	13	41	0.585	4.7	−109.6
			fclust	60	24	36	0.982	40.4	−236.9
		42 days	lm	55	7	47	0.557	8.8	−280.6
			fclust	60	10	50	0.540	4.2	−269.9
	priming effect	7 days	lm	55	24	30	0.724	3.5	−293.1
			fclust	60	17	43	0.937	23.3	−412.7
		42 days	lm	55	23	31	0.848	8.0	−304.2
			fclust	60	8	52	0.838	20.5	−361.0
bacteria and fungi	soil	7 days	fclust	120	13	107	0.949	181.4	−320.0
		42 days	fclust	120	14	106	0.971	59.3	−358.2
	straw	7 days	fclust	120	14	106	0.938	37.1	−214.7
		42 days	fclust	120	20	100	0.979	29.2	−407.4
	Priming effect	7 days	fclust	120	19	101	0.959	49.3	−428.1
		42 days	fclust	120	13	107	0.933	145.1	−395.7

Coefficient of determination (R^2^), Fisher ratio (F-ratio), and Akaike Information Criterion corrected for small samples (AICc) of the models.

**Table 2 microorganisms-11-01106-t002:** Bacterial and fungal families identified **separately** by multi-linear (lm) and clustering analyses (fclust) as key actors in basal soil respiration (soil), straw mineralization (straw), and the priming effect.

Mineralisation	Incubation Time	Model	Bacteria	*Unknown*	*Planctomycetaceae*	*Acidobacteria_Gp1*	*Acidobacteria_Gp2*	*Thermomonosporaceae*	*Ktedonobacteraceae*	*Undefined*	*Bradyrhizobiaceae*	*Acidobacteria_Gp4*	*Hyphomicrobiaceae*	*Acetobacteraceae*	*Chitinophagaceae*	*Conexibacteraceae*	*Cystobacteraceae*	*Mycobacteriaceae*	*Pseudonocardiaceae*	*Burkholderiaceae*	*Oxalobacteraceae*	*Polyangiaceae*	*Gemmatimonadaceae*	*Chloroplast*	*Acidobacteria_Gp6*	*Chthonomonadaceae*	*Acidobacteria_Gp7*	*Streptomycetaceae*	*Rhodospirillaceae*	*Micromonosporaceae*	*Fervidicoccaceae*	*Acidimicrobineae_incertae_sedis*	*Paenibacillaceae*	*Xanthomonadaceae*	*Beijerinckiaceae*	*Bacillaceae*	*Comamonadaceae*	*Catenulisporaceae*	*Nocardioidaceae*	*Caulobacteraceae*	*Actinospicaceae*	*Acidobacteria_Gp13*	*Bdellovibrionaceae*	*Solirubrobacteraceae*	*Acidobacteria_Gp5*	*Rubrobacteraceae*	*Haliangiaceae*	*Planococcaceae*	*Sinobacteraceae*	*Coxiellaceae*	*Cyanobacteria.Chloroplast_F1*	*Methylocystaceae*	*Geodermatophilaceae*	*Rhizobiales_incertae_sedis*	*Armatimonadaceae*	*Acidobacteria_Gp3*	*Xanthobacteraceae*	*Flavobacteriaceae*	*Pasteuriaceae*	*Phyllobacteriaceae*	*Methylobacteriaceae*	*Nitrospiraceae*	*Microbacteriaceae*
soil	7 d	lm		***	***	^.^	***	***			***	^.^	***		***				^.^		***	***	^.^		^.^	^.^	^.^		^.^		^.^	^.^		^.^	^.^	^.^		***				***						^.^	***	^.^	^.^	***		^.^		***		^.^					
				−	−		−	−			+		−		−						−	+																−				−							−			+				+							
		fclust							***		***		***																									***														***				***							
				D				E	C−		A+		C−		E		E									E				E							D	B+							E			D			E	A+				C−	D						
	42 d	lm				***	***		***			***	^.^	***		^.^	^.^			***	***	***			***				^.^		***	^.^	^.^	^.^			***	***							^.^	^.^	^.^	^.^			^.^	***		^.^		^.^		***					
						−	−		−			−		−						+	−	+			+						−						−	−														+						−					
		fclust								***	***											***																				***	***		***					***													
										B+	A+		E		C							B+												C				E				D+	D+		A+					D+						C							
straw	7 d	lm			^.^	^.^	***	***	***	***	^.^	***	***	^.^		***				***		^.^	^.^		***	^.^	***		***	^.^	^.^	^.^			^.^	^.^	^.^		***	^.^	^.^				^.^	***	^.^	***	***	^.^	^.^	^.^		***	***		***						
							−	+	−	−		−	+			+				+					+		−		+										+							−		−	+					−	+		+						
		fclust											***	***																						***			***			***				***											***						
										E			B−	B−												E			C							B−			A+		D	B−			D	B−											A+						C
	42 d	lm		***	^.^	***		***	***						^.^		***	^.^		^.^				^.^		^.^					^.^		^.^					^.^	^.^	***		^.^								^.^				^.^									
				−		−		+	+								−																							+																							
		fclust		***				***				***	***			***														***											***										***												
				A+				A+				B+	A+			B+		E						C					C	A+				E			D				A+										A+	D										D	
priming effect	7 d	lm		***	^.^	^.^		^.^		^.^		^.^				^.^	^.^		^.^	^.^		^.^			^.^	^.^	^.^							^.^		***	^.^	***				***							***						^.^	^.^							
				+																																+		+				+							+														
		fclust		***	***				***							***						***													***	***				***	***											***	***		***						***		
				B−	A+			D	B−					F		C−						A+				E						E			B−	A+				A+	C−		D		F							A+	A+		B−			E			C−		
	42 d	lm			***		^.^		***		^.^	^.^	^.^	^.^	***		***			***				^.^	***	^.^		^.^			***	***					***	^.^				^.^				^.^	^.^	^.^		^.^		^.^											
					−				+						−		−			−					−						+	−					+																										
		fclust		***											***							***								***										***							***																
				A+											D+			E				A+		E						C+			E				E			D+							D+				B		E	B									
			**Fungi**	*Unclassified*	*Unknown*	*Tricholomataceae*	*Environmental*	*Glomeraceae*	*Herpotrichiellaceae*	*Mortierellaceae*	*Tubeufiaceae*	*Phaeosphaeriaceae*	*Myxotrichaceae*	*Trichocomaceae*	*Chaetomiaceae*	*Coniophoraceae*	*Teratosphaeriaceae*	*Davidiellaceae*	*Nectriaceae*	*Strophariaceae*	*Endogonaceae*	*Lyophyllaceae*	*Hypocreaceae*	*Pleurotaceae*	*Bulgariaceae*	*Mycosphaerellaceae*	*Entolomataceae*	*Pyronemataceae*	*Pluteaceae*	*Boletaceae*	*Helotiaceae*	*Tremellaceae*	*Coniochaetaceae*	*Cordycipitaceae*	*Hyaloscyphaceae*	*Clavicipitaceae*	*Pleosporaceae*	*Lycoperdaceae*	*Gomphaceae*	*Marasmiaceae*	*Corticiaceae*	*Monoblepharidaceae*	*Stereaceae*	*Plectosphaerellaceae*	*Magnaporthaceae*	*Ajellomycetaceae*	*Didymellaceae*	*Agaricaceae*	*Massariaceae*	*Agyriaceae*	*Orbiliaceae*	*Pezizaceae*	*Ophiocordycipitaceae*	*Montagnulaceae*	*Rhizophydiaceae*	*Trechisporaceae*	*Ophiostomataceae*	*Kickxellaceae*	*Bionectriaceae*	*Sarcosomataceae*	*Ambisporaceae*	*Ustilaginaceae*	*Auriculariaceae*
soil	7 d	lm								^.^												^.^				^.^					^.^					^.^				^.^					^.^				^.^				***		^.^	^.^							
																																																					+										
		fclust		^.^	^.^		^.^								^.^	^.^	^.^	^.^	^.^			***				***	^.^					^.^	^.^		^.^		^.^								^.^			***	***	***		^.^		***		^.^		^.^	***	^.^	^.^		
				F	F		F								F	F	F	F	F			A+				A+	D					F	F		F		E								D			C	B+	A+		F		B+		F		E	B+	E	D		
	42 d	lm								^.^	^.^	^.^			^.^		^.^	^.^				^.^		^.^	^.^	^.^		^.^			^.^					^.^	^.^		^.^	^.^				^.^	^.^	^.^		^.^	***				***					***					
																																																	+				+					+					
		fclust		^.^							***				***												^.^				^.^							***							^.^	^.^	^.^			***									^.^	***	***	^.^	
				E							D−				B+												E				E							A+							C	E	C			A+									E	B+	D−	C	
straw	7 d	lm				^.^			^.^											^.^		^.^			^.^		***							^.^						^.^			^.^										^.^			^.^	^.^	^.^					
																											−																																				
		fclust						***	***	^.^	***	***		***								^.^						^.^	***		^.^		***									^.^		^.^	^.^		***		^.^		^.^	^.^		^.^			^.^	^.^			***	***	^.^
								B−	C+	F	B−	C+		B−								F						D	A+		D		C+									F		F	E		B−		D		F	F		E			F	F			A+	A+	F
	42 d	lm										***	^.^												^.^	^.^								***																^.^								^.^					
												−																						−																													
		fclust																			***				^.^			***	***		***			***				^.^																	***				^.^	^.^			
																					A−				B			A−	A−		A−			A−				B																	A−				B	B			
priming effect	7d	lm			^.^			^.^	^.^			^.^			^.^		^.^		^.^	^.^	^.^		^.^	^.^			^.^	^.^					^.^				^.^			^.^	^.^	^.^						^.^	^.^		***		^.^				^.^	***					
																																																			+							+					
		fclust		***				^.^		***			***			^.^	***													^.^							^.^					***	^.^	^.^		***			***						***				^.^	***		***	
				A+				E		A+			A+			E	A+													C							C					A+	C	E		D−			B						D−				C	B−		B−	
	42 d	lm							^.^					^.^	^.^		^.^		^.^	^.^			^.^		^.^	***	^.^	^.^		^.^	^.^	^.^							^.^	^.^	***	^.^			^.^			^.^				^.^					^.^	***					
																										+															−																	−					
		fclust										***		^.^	^.^					^.^	^.^							^.^																		^.^								***									
												B+		A	C					D	A							D																		A								B+									

A midpoint (·) indicates that the family was identified as key by the variable selection procedure and (***) indicates that the significance was at *p* < 0.001. Cells are colored depending on if the family is identified as key by (pink) both the multi-linear and clustering models, (yellow) the multi-linear model alone, (green) the clustering model alone, (blue) as negligible by both the multi-linear and clustering models, or (white) as not analyzed by the multi-linear model. The sign indicates the direction, positive (+) or negative (−), of the family’s contribution to SOM decomposition activity. For clustering analysis, the letter (A to F) indicates the most significant functional group to which a class of relative abundance of the family was assigned. Families were ranked from left to right in descending order of relative abundance.

**Table 3 microorganisms-11-01106-t003:** Bacterial and fungal families identified **altogether** by clustering analyses (fclust) as key actors in basal soil respiration (soil), straw mineralization (straw), and the priming effect.

Mineralisation	Incubation Time	Model	Bacteria	*Unknown*	*B.Planctomycetaceae*	*B.Acidobacteria-Gp1*	*B.Acidobacteria-Gp2*	*B.Thermomonosporaceae*	*B.Ktedonobacteraceae*	*B.Undefined*	*B.Bradyrhizobiaceae*	*B.Acidobacteria-Gp4*	*B.Hyphomicrobiaceae*	*B.Acetobacteraceae*	*B.Chitinophagaceae*	*B.Conexibacteraceae*	*B.Cystobacteraceae*	*B.Mycobacteriaceae*	*B.Pseudonocardiaceae*	*B.Burkholderiaceae*	*B.Oxalobacteraceae*	*B.Polyangiaceae*	*B.Gemmatimonadaceae*	*B.Chloroplast*	*B.Acidobacteria-Gp6*	*B.Chthonomonadaceae*	*B.Acidobacteria-Gp7*	*B.Streptomycetaceae*	*B.Rhodospirillaceae*	*B.Micromonosporaceae*	*B.Fervidicoccaceae*	*B.Acidimicrobineae-incertae-sedis*	*B.Paenibacillaceae*	*B.Xanthomonadaceae*	*B.Beijerinckiaceae*	*B.Bacillaceae*	*B.Comamonadaceae*	*B.Catenulisporaceae*	*B.Nocardioidaceae*	*B.Caulobacteraceae*	*B.Actinospicaceae*	*B.Acidobacteria-Gp13*	*B.Bdellovibrionaceae*	*B.Solirubrobacteraceae*	*B.Acidobacteria-Gp5*	*B.Rubrobacteraceae*	*B.Haliangiaceae*	*B.Planococcaceae*	*B.Sinobacteraceae*	*B.Coxiellaceae*	*B.Cyanobacteria.Chloroplast-F1*	*B.Methylocystaceae*	*B.Geodermatophilaceae*	*B.Rhizobiales-incertae-sedis*	*B.Armatimonadaceae*	*B.Acidobacteria-Gp3*	*B.Xanthobacteraceae*	*B.Flavobacteriaceae*	*B.Pasteuriaceae*	*B.Phyllobacteriaceae*	*B.Methylobacteriaceae*	*B.Nitrospiraceae*	*B.Microbacteriaceae*
soil	7 d	fclust									***														∙		∙				***							***	^.^											∙	∙	***											
											A+														E		D				B+							B+	C+											D	E	A+											
	42 d	fclust									***							∙		∙		***											∙					***							***	***																	
											A+							D		D		B+											E					C−							A+	B+																	
straw	7 d	fclust											***	***														∙						∙	∙				***																	***	***				∙		
													B−	B−														D						E	C				A+																	B−	A+				C		
	42 d	fclust		***				***					***							***		∙		***			***		∙						***	∙		***			***					∙							***										
				A+				A+					A+							D−		E		D−			C+		F						B+	E		C+			A+					E							D−										
priming effect	7 d	fclust					***																												∙	***				***				***	***			***				***				***	***			***			
							D−																												E	A+				A+				D−	B+			C−				A+				C−	C−			B+			
	42 d	fclust		***																***		***							***												***								***											***			
				A+																C−		A+							C−												A+								C−											C−			
			**Fungi**	*F.Unclassified*	*F.Unknown*	*F.Tricholomataceae*	*F.Environmental*	*F.Glomeraceae*	*F.Herpotrichiellaceae*	*F.Mortierellaceae*	*F.Tubeufiaceae*	*F.Phaeosphaeriaceae*	*F.Myxotrichaceae*	*F.Trichocomaceae*	*F.Chaetomiaceae*	*F.Coniophoraceae*	*F.Teratosphaeriaceae*	*F.Davidiellaceae*	*F.Nectriaceae*	*F.Strophariaceae*	*F.Endogonaceae*	*F.Lyophyllaceae*	*F.Hypocreaceae*	*F.Pleurotaceae*	*F.Bulgariaceae*	*F.Mycosphaerellaceae*	*F.Entolomataceae*	*F.Pyronemataceae*	*F.Pluteaceae*	*F.Boletaceae*	*F.Helotiaceae*	*F.Tremellaceae*	*F.Coniochaetaceae*	*F.Cordycipitaceae*	*F.Hyaloscyphaceae*	*F.Clavicipitaceae*	*F.Pleosporaceae*	*F.Lycoperdaceae*	*F.Gomphaceae*	*F.Marasmiaceae*	*F.Corticiaceae*	*F.Monoblepharidaceae*	*F.Stereaceae*	*F.Plectosphaerellaceae*	*F.Magnaporthaceae*	*F.Ajellomycetaceae*	*F.Didymellaceae*	*F.Agaricaceae*	*F.Massariaceae*	*F.Agyriaceae*	*F.Orbiliaceae*	*F.Pezizaceae*	*F.Ophiocordycipitaceae*	*F.Montagnulaceae*	*F.Rhizophydiaceae*	*F.Trechisporaceae*	*F.Ophiostomataceae*	*F.Kickxellaceae*	*F.Bionectriaceae*	*F.Sarcosomataceae*	*F.Ambisporaceae*	*F.Ustilaginaceae*	*F.Auriculariaceae*
soil	7 d	fclust					***																	^.^																													^.^		^.^								
							B+																	D																													D		C								
	42 d	fclust							^.^		***			***		***							^.^																																		***						
									E		C−			B+		C−							E																																		C−						
straw	7 d	fclust		^.^							***																^.^															***					***																
				E							B−																D															A+					B−																
	42 d	fclust									^.^																													***						***							***									***	***
											F																													B+						C+							A+									B+	B+
priming effect	7 d	fclust									***						***											***	***					***													^.^													***	***		
											C−						A+											A+	C−					D−													E													A+	B+		
	42 d	fclust										***			***		^.^																													***						^.^		***									
												A+			B+		E																													A+						E		A+									

A midpoint (·) indicates that the family was identified as key by the variable selection procedure and (***) indicates that the significance was at *p* < 0.001. Cells are colored depending on if the family is identified as (green or pink) key by the clustering model or as (blue) negligible. Cells colored in pink were also previously identified as key by the multi-linear model performed on separated bacterial and fungal communities’ analyses (see Table 2). The sign indicates the direction, positive (+) or negative (−), of the family’s contribution to SOM decomposition activity. The letter (A to F) indicates the most significant functional group to which a class of relative abundance of the family was assigned. Families were ranked from left to right in descending order of relative abundance.

## Data Availability

DNA sequences were deposited in the European Nucleotide Archive, under the study accession number PRJEB19651. The package functClust is available on R-CRAN at https://CRAN.R-project.org/package=functClust (accessed on 20 March 2023). The datasets and R-codes used will be available on the INRAE dataverse at https://doi.org/10.15454/LRY3M2 (accessed on 20 March 2023).

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
