# Peer review of "Nonlinear Effects Induced by Interactions among Functional Groups of Bacteria and Fungi Regulate the Priming Effect in Malagasy Soils"

_microorganisms, 2023, doi:10.3390/microorganisms11051106_

Round 1

Reviewer 1 Report

The article describes interesting findings based on the taxonomic composition analysis of soil microbial community and its relation to SOM decomposition. The tested hypothesis is described quite clear, and the conclusions answer the stated questions. However, there are some moments that should be corrected.

Abstract, and also materials and methods section, could be improved by a bit more detailed description of experiment. Why the 7th and 42nd days of incubation were chosen? What were soil properties on different sites?

Bioinformatic analysis should also be described in more detail.

Also, is it possible to define the Unclassified families or exclude them from the analysis? Even though the abundance of families “Unknown, Uclassified, Undefined” are relatively high, maybe it’s not a good idea to include them in the study, because it could be a mixed group of different taxonomic affiliation. This could be fixed on the step of bioinformatic data treatment, when ASVs are picked instead of OTUs and analyzed further.

If this problem couldn't be fixed, than I suggest some discussion of this limitation in the Discussion section

Lines 15-20 - please clarify, are there two theories or two processes happening at the same time. Processes could depend on the microbial population properties, while theories could only be explained by some of these properties and processes. Same for Lines 45-50.

Line 103 – how the priming effect was estimated? The other two properties are quite clear, while the priming effect needs more explanation at this point

Line 114 – what was the difference between these 57 sites? I see that there is a link to another article regarding these samples, but it would be nice to have a brief description what factors were different (m.b. type of plants on sites, or parental rock, or soil types?) As could be seen from results, the difference between sampling sites was important and affected the SOM decomposition

Line 126 – what version of SILVA was used? Was the bioinformatic treatment performed again to define the sequences affiliations to exact families using fresher version of SILVA database?

Lines 151, 158, 164, 187, 212, 224, 233, 243 – please consider adding sub-sections here, or paraphrase the first sentences to improve the readability.

Lines 182-185 – why did you choose the number of families as a threshold, but not their abundances?

Line 308 - Table 2, should it be named “figure”? At the heading, “substrate” should be corrected (it’s not a substrate, it’s a measured process). Also, letters A-D are not explained in the description. Same for Table 3 (and also there is no yellow color that was mentioned in the description)

Line 368, Figure 1 – are the colors meaningful? Maybe they should be described.

Line 437 – class2/2 – please check if it is a mistake, in the methods section it was written that the classes are 1/3,2/3,3/3 and 1/1

Lines 465-467 – repetition of the 1st section name

Reviewer 2 Report

The manuscript matches the journal profile.The aim of the study was to identify the microbial taxa that predominantly contribute to the generation of the priming effect (PE) in Madagascar ferralsols, either through their own actions or through their interactions with other microbial taxa.

The research was well done methodologically. The abstract as well as the introduction and discussion chapters are well written. In the Results chapter, I suggest shortening the titles of Table 2 and Table 3, and placing the necessary explanations under the tables. The results presented in table 2 are illegible. There is also no separate chapter on conclusions.

Round 2

Reviewer 1 Report

Dear authors, thank you for taking my comments into consideration. As I see, the article is improved now. Introduction becomes clearer. However, from my point of view, it would be nice to leave there the descriptions of "stoichiometric decomposition” and “nutrient mining”. 

Line 17 – in my opinion, decomposition abilities sounds better 

Lines 517-518 – it would be good if you add here some description why Unknown appeared in the dataset. I mean, one can’t seriously describe ecological features of Unknown bacteria; there could be several different taxa with contrasting properties that were sorted in this group

Author Response

Comments and Suggestions for Authors

Dear authors, thank you for taking my comments into consideration. As I see, the article is improved now. Introduction becomes clearer. However, from my point of view, it would be nice to leave there the descriptions of "stoichiometric decomposition” and “nutrient mining”. 

We agree and have corrected the abstract

Line 17 – in my opinion, decomposition abilities sounds better 

Done

Lines 517-518 – it would be good if you add here some description why Unknown appeared in the dataset. I mean, one can’t seriously describe ecological features of Unknown bacteria; there could be several different taxa with contrasting properties that were sorted in this group

We agree with the reviewer and have modified the discussion section Lines 521-524 and 614-617.